# Mortality Rate and Predictors of Mortality in Hospitalized COVID-19 Patients with Diabetes

**DOI:** 10.3390/healthcare8030338

**Published:** 2020-09-13

**Authors:** Dilaram Acharya, Kwan Lee, Dong Seok Lee, Yun Sik Lee, Seong-Su Moon

**Affiliations:** 1Department of Preventive Medicine, College of Medicine, Dongguk University, Gyeongju 38066, Korea; dilaramacharya123@gmail.com (D.A.); kwaniya@dongguk.ac.kr (K.L.); 2Department of Community Medicine, Kathmandu University, Devdaha Medical College and Research Institute, Rupandehi 32900, Nepal; 3Department of Pediatrics, Dongguk University College of Medicine, Gyeongju 38067, Korea; lds117@dongguk.ac.kr; 4Department of General Surgery, Andong Medical Center, Andong 36694, Korea; ysleewmbh@hanmail.net; 5Department of Internal Medicine, Dongguk University College of Medicine, Dongguk University Gyeongju hospital, Dongdae-ro 87, Gyeongju 38067, Korea

**Keywords:** coronavirus, diabetes, mortality, pandemics, Korea

## Abstract

Studies have confirmed COVID-19 patients with diabetes are at higher risk of mortality than their non-diabetic counterparts. However, data-driven evidence of factors associated with increased mortality risk among hospitalized COVID-19 patients with diabetes is scarce in South Korea. This study was conducted to determine the mortality rate and identify risk factors of mortality among hospitalized COVID-19 patients with type 2 diabetes in Gyeongsangbuk-do province, South Korea. In this hospital-based, cross-sectional study, we enrolled a total of 324 patients with confirmed COVID-19, hospitalized at two of the tertiary level healthcare facilitates of Gyeongsangbuk-do, South Korea from 18 February to 30 June 2020. Demographic and clinical data and laboratory profiles were analyzed and multivariate logistic regression analysis was used to identify risk factors of mortality among diabetic patients with COVID-19. Of the 324 patients, 55 (16.97%) had diabetes mellitus. The mean age of all study subjects was 55 years, and the mean age of those with diabetes was greater than that of those without (69.8 years vs. 51.9 years). Remarkably, the mortality rate was much higher among those with diabetes (20.0% vs. 4.8%). Multivariate logistic regression analysis revealed that an older age (≥70 years) and a high serum lactate dehydrogenase (LDH) levels significantly predicted mortality among hospitalized COVID-19 patients with diabetes. Our study cautions more attention to be paid to patients with diabetes mellitus hospitalized for COVID-19, especially those aged ≥ 70 years and those with a high serum LDH level, to reduce the risk of mortality.

## 1. Introduction

Severe acute respiratory syndrome coronavirus-2 (SARS-CoV-2) first emerged in Wuhan, China, in late 2019 and has since reached pandemic proportions, caused 646,000 deaths, and resulted in more than 16 million cases of COVID-19 as of 27 July 2020 [1]. In South Korea, 13,007 confirmed cases, and a total of 300 deaths were reported as of 28 July [2]. In addition to its devastating effects on health, the virus has caused severe economic disruption, and thus, has markedly reduced quality of life worldwide [3,4,5,6]. It would seem that the COVID-19 pandemic is set to continue until game-changers like an effective vaccine and or therapy appear [7]. Infections are difficult to manage, and mortality rates are likely to increase until a viable vaccine or chemotherapeutic becomes available. Until then we are obliged to rely on preventive measures that have proven to be effective, such as wearing masks, washing hands frequently, maintaining social distance, contact tracing, and testing [6,8,9].

Comorbid conditions such as diabetes mellitus, respiratory, cardiovascular and renal illnesses, and obesity, and an older age group are known to be positively associated with poor outcomes among COVID-19 patients [10,11,12,13]. Furthermore, research studies have demonstrated that COVID-19 patients with diabetes have poorer clinical prognoses, are at higher risks of complications, such as respiratory failure and acute cardiac injury, and have shorter overall survival times than their non-diabetic counterparts [14].

Several systematic reviews and meta-analyses [15,16], cohort [14,17], case-control [12], and cross-sectional studies [16,18] have demonstrated that diabetic patients with COVID-19 are at considerably greater risk of mortality and of developing more severe disease than COVID-19 patients without diabetes. However, it was reported in a recent study that diabetes had no effect on the prognosis of COVID-19 patients but was negatively associated with clinical outcomes [19], which indicated studies are needed to narrow gaps in our understanding of the disease, and thus, aid decision-making by health care providers and administrators. Given the current level of understanding of COVID-19 knowledge, we undertook to determine the additional risk posed by COVID19 to diabetic patients, to document mortality rates, and identify risk factors of mortality among COVID-19 patients with type 2 diabetes hospitalized in Gyeongsangbuk-do province, South Korea.

## 2. Materials and Methods

### 2.1. Study Design and Subjects

Three hundred and twenty-four patients (135 males and 189 females) with a diagnosis of COVID-19, hospitalized in Dongguk University Gyeongju Hospital (DUGH) or Andong Medical Center (AMC), Gyeongsangbuk-do, Korea from 18 February to 30 June 2020, were enrolled in this cross-sectional study. Patients re-hospitalized due to a positive RT-PCR result after discharge or were still in hospital were excluded. Only initial AMC records of patients transferred from AMC to DUGH were included. Demographic and clinical characteristics, including chest radiologic (X-ray or computerized tomography (CT) and laboratory findings, were collated. Laboratory blood analysis was performed at time of admission and included blood glucose, complete blood cell (CBC) counts, aspartate aminotransferase (AST), alanine aminotransferase (ALT), glycosylated hemoglobin (HbA1c), lactate dehydrogenase (LDH), blood urea nitrogen (BUN) and creatinine levels, albumin, and C-reactive protein (CRP). Therapeutics administered included lopinavir/ritonavir, chloroquine, antibiotics, macrolide, pegylated interferon-α, and oxygen supplementation.

### 2.2. Statistical Analysis

Nominal variables are presented as numbers of cases and percentages, and continuous variables as means and standard deviations (SDs). The student’s *t*-test or the Mann–Whitney U test was used to determine the significances of differences between two groups (Diabetes vs. Non-diabetes), and Chi-square analysis or Fisher’s exact test was used to determine the significances of differences between categorical variables. Deaths included patients that had expired after transfer to another hospital. Laboratory parameters were classified as ‘above normal limit’, ‘normal’ or ‘below normal limit’ based on reference ranges, as follows; high AST, >39 U/L (male) and >31 U/L (female); high ALT, >40 U/L (male) and >32 U/L (female); high LDH, >225 U/L (male) and >214 U/L (female); high creatinine, 1.3 mg/dL (male) and 1.1 mg/dL (female); anemia, <14 g/dL (male) and <12 mg/dL (female); low albumin, <3.5 mg/dL; high CRP, >0.5 mg/dL. Univariate logistic regression analysis (adjusted for age) was used to investigate associations between mortality risk and clinical or laboratory parameters. LDH and age ≥ 70 years were included as potential risk factors included in the univariate analysis. Multivariate logistic regression analysis was used to investigate the natures of associations between potential risk factors identified by univariate analysis and mortality. All tests were two-sided, and *p*-values of <0.05 were considered to indicate statistical significance. The analysis was conducted using the Statistical Package for Social Science Ver. 20.0 (SPSS, Chicago, IL, USA).

### 2.3. Ethical Statement

This study protocol was exempted from ethical review by the Institutional Review Board of Dongguk University Gyeongju Hospital because the data analyzed did not contain information that could be used to identify individual patients (IRB registration No. 110757-202006-HR-04-02).

## 3. Results

### 3.1. Clinical Characteristics of the Study Subjects

The baseline characteristics of the 324 study subjects are provided in Table 1 and Table 2. Mean subject age was 55 years, and 55 had type 2 diabetes. The mean age in the diabetes group was greater than in the non-diabetes group (69.8 years vs. 51.9 years). The mortality rate in the diabetes group was remarkably higher than in the non-diabetes group (20.0% vs. 4.8%). Patients in the diabetes group stayed in hospital longer (22.8 vs. 18.5 days. *p* = 0.035) and had significantly higher comorbidity rates (e.g., hypertension, dyslipidemia, cardiovascular disease, and dementia) (*p* < 0.05). Chest radiology showed bilateral pneumonia was more common in the diabetes group (67.3% vs. 43.1%). The laboratory findings showed blood glucose levels were higher and that AST or CRP elevation, anemia, and a low albumin level were more frequent in the diabetes group. Furthermore, more drugs were administered to patients with diabetes than to those without.

### 3.2. Risk Factors of Mortality in COVID-19 Patients with Diabetes

Univariate regression analysis showed an age of ≥70 years and elevated LDH were significantly related to mortality in the diabetes group. However, other parameters including body mass index, smoking, alcohol drinking, leukocytosis, lymphopenia, glucose, and HbA1c were not significantly related to mortality (Table 3). Multivariate regression analysis showed an age of ≥70 years and high LDH were significantly independent predictors of mortality among COVID-19 patients with diabetes (Table 4).

## 4. Discussion

In this era defined by the COVID-19 pandemic, early identification and reduction of mortality in those at risk population are essential, especially among COVID-19 patients with comorbid conditions, such as diabetes, cardiovascular disorders, renal diseases, and malignancies, to reduce mortality and minimize pain and suffering. This pandemic has suggested that the management of vulnerable populations might be successful when data-driven policies and strategies are adopted.

In the present study, we found that among 324 confirmed hospitalized patients with COVID-19, 55 (16.97%) had diabetes mellitus, and that the mean age of patients with diabetes was greater than those without (69.8 years vs. 51.9 years). Notably, we found that the mortality rate was much higher among diabetic patients than in those without (20.0% vs. 4.8%). Other authors have also reported higher rates of comorbid conditions such as cardiovascular, respiratory illnesses, chronic kidney diseases, malignancies, and diabetes mellitus in hospitalized COVID-19 patients [20,21,22,23], and diabetes was found to account for most of these comorbid conditions [24]. Moreover, the presence of diabetes in COVID-19 patients has been consistently reported to increase disease severity and the mortality risk [25,26]. A recent Chinese study reported that diabetes was an associated comorbidity in 14% of patients that survived COVID-19, but in 31% of those that did not survive [25], and Shenoy et al., reported that the prevalence of diabetes was almost three times higher in COVID-19 patients with severe disease (16.2%) than in those with non-severe disease (5.7%) [26]. These findings suggest COVID-19 patients with diabetes should be prioritized and that focused preventive strategies be instituted to address diabetes and other associated comorbid conditions in order to reduce growing COVID-19 mortality and other poor health outcomes.

Interestingly, multivariate logistic regression analysis revealed that an age of ≥70 years and a high serum LDH level significantly predicted mortality among hospitalized COVID-19 patients with diabetes. Similarly, other studies have reported that an age of >60 years is a major predictor of disease severity and mortality among diabetic COVID-19 patients [16,17]. A growing body of literatures supported an age of ≥70 years’ patients with COVID-19 had incremental severity and mortality than younger ones [23,27]. For example, a recent modeling study estimated that 20% of those aged 70 years or older patients with COIVD-19 were severe requiring hospitalization whereas only <1% of those younger than 20 years needed so [23], while a Korean study reported that the age-specific death rate was the highest among patients over 70 years of age, with underlying diseases in their circulatory system [27]. Older individuals with diabetes mellitus and an acute respiratory illness are usually susceptible to develop complications such as severe acute respiratory syndrome (SARS), Middle East Respiratory Syndrome (MERS), influenza (including COVID-19), and thus, to be at higher risk of mortality [10]. Furthermore, it has been suggested that diabetic COVID-19 patients are more susceptible to an ‘inflammatory storm’, which is associated with rapid deterioration and high mortality risk [18].

As regards LDH levels, studies have reported poor clinical outcomes for patients with other viral infections when LDH levels are elevated, and as a result, LDH is considered an important biomarker of mortality [28,29]. In our study the cut-off value of LHD was set at 225 and 214 for male and female based on existing practices in South Korea in order to identify the associated risk for COVID-19 mortality. The differential in cut-off measurements might have varied reported results. Nonetheless, a meta-analysis identified significant differences between the LDH levels of COVID-19 patients with severe diseases and those without [30]. Likewise, another recent pooled analysis confirmed that LDH is positively associated with poorer outcomes in COVID-19 patients, and found that a high LDH level was associated with a 6-fold increase in the risk of developing severe disease and a 16-fold increase in mortality risk [31].

Our study findings should be interpreted in the light of certain specific limitations: First, identified predictors in Korean subjects might not be applicable to all human races. Second, the study was performed on a relatively small number of hospitalized COVID-19 patients at two hospitals, which may limit the generalizability of our findings to other settings. However, we believe that our results are meaningful as they should aid the development of customized preventive strategies aimed at reducing the burden of COVID-19. Additional large-scale studies are required to determine mortality and disease severity risks among hospitalized COVID-19 patients with other associated comorbid conditions.

## 5. Conclusions

In conclusion, our findings show that greater attention should be directed toward COVID-19 patients with diabetes older than 70 years and with those having an elevated LDH level to reduce the risk of mortality, and that focused preventive strategies be devised and implemented in this patient population. Further studies should be conducted in a large sample size in order to increase external validity and methods could rule out limitations inherent to this study.

## Figures and Tables

**Table 1 healthcare-08-00338-t001:** General characteristics of the study subjects.

Variables	Total Subjects	Diabetes	Non-Diabetes	*p*-Value
Number of cases	324	55	269	
Age, years	55.0 ± 21.4	69.8 ± 13.5	51.9 ± 21.4	<0.001
Body Mass Index, kg/m^2^	23.7 ± 4.4	24.5 ± 4.9	23.5 ± 4.3	0.151
Systolic blood pressure, mmHg	122.2 ± 20.0	127.5 ± 25.2	121.1 ± 18.7	0.032
Diastolic blood pressure, mmHg	77.6 ± 12.6	79.5 ± 16.2	77.3 ± 11.7	0.241
Body temperature	36.6 ± 1.9	36.7 ± 0.9	36.6 ± 2.0	0.787
Sex				0.454
Male	135 (41.7)	20 (36.4)	115 (42.8)	
Female	189 (58.3)	35 (63.6)	154 (57.2)	
Current smoker	55 (17.0)	11 (20.0)	44 (16.4)	0.554
Alcohol drinker	52 (16.0)	10 (18.2)	42 (15.6)	0.687
Resident in care facility	87 (26.9)	22 (40.7)	65 (24.2)	0.018
Days of hospitalization	19.2 ± 13.6	22.8 ± 18.1	18.5 ± 12.3	0.035
Death	24 (7.4)	11 (20.0)	13 (4.8)	0.001
Comorbidity				
Hypertension	80 (24.7)	32 (58.2)	48 (17.8)	<0.001
Dyslipidemia	25 (7.7)	9 (16.4)	16 (5.9)	0.022
Cardiovascular disease	19 (5.9)	8 (14.6)	11 (4.1)	0.009
Cerebrovascular disease	21 (6.5)	7 (12.7)	14 (5.2)	0.064
Dementia	29 (9.0)	10 (18.2)	19 (7.1)	0.016
Malignancy	13 (4.0)	3 (5.5)	10 (3.7)	0.469
Symptoms on admission				
asymptomatic	67 (20.7)	12 (21.8)	55 (20.4)	0.855
Cough	105 (32.4)	10 (18.2)	95 (35.3)	0.017
Sputum	57 (17.6)	8 (14.5)	49 (18.2)	0.697
Rhinorrhea	12 (3.7)	2 (3.6)	10 (3.7)	0.999
Sore throat	52 (16.0)	4 (7.3)	48 (17.8)	0.068
Dyspnea	31 (9.6)	12 (21.8)	19 (7.1)	0.002
Gustatory dysfunction	4 (1.2)	0	4 (1.5)	0.999
Olfactory dysfunction	10 (3.1)	0	10 (3.7)	0.222
Febrile sense	94 (29.0)	17 (30.9)	77 (28.6)	0.746
Headache	42 (13.0)	5 (9.1)	37 (13.8)	0.508
Myalgia	43 (13.3)	4 (7.3)	39 (14.5)	0.192
Fatigue	16 (4.9)	3 (5.5)	13 (4.8)	0.741
Chest pain/discomfort	25 (7.7)	5 (9.1)	20 (7.4)	0.591
Nausea/vomiting	3 (0.9)	1 (1.8)	3 (1.1)	0.999
Diarrhea	13 (4.0)	3 (5.5)	10 (3.7)	0.469

Data are shown as mean ± standard deviation or frequency (%). *p*-value for differences between diabetes and non-diabetes.

**Table 2 healthcare-08-00338-t002:** Chest imaging findings, laboratory characteristics, and treatments.

Variables	Total Subjects	Diabetes	Non-Diabetes	*p*-Value
Chest imaging				0.001
None	134 (41.4)	10 (18.2)	124 (46.1)	
Bilateral infiltration	153 (47.2)	37 (67.3)	116 (43.1)	
Unilateral infiltration	37 (11.4)	8 (14.5)	29 (10.8)	
Laboratory findings				
Blood glucose, mg/dL	123.7 ± 51.7	189.1 ± 90.0	110.2 ± 22.6	<0.001
HbA1c, %	7.27 ± 1.89	8.08 ± 2.07	6.14 ± 0.69	0.001
Leukocyte, >10.7 × 10^9^/L	29 (5.9)	5 (9.1)	14 (5.2)	0.339
Lymphocyte, <1.0 × 10^9^/L	62 (19.1)	15 (27.3)	47 (17.5)	0.131
High Aspartate aminotransferase	85 (26.2)	16 (29.1)	69 (25.7)	0.615
High Alanine aminotransferase	53 (16.4)	8 (14.5)	45 (16.7)	0.842
High Lactate dehydrogenase	62 (19.1)	19 (34.5)	43 (16.0)	0.003
High Creatinine	22 (6.8)	7 (12.7)	15 (5.6)	0.073
Anemia	72 (22.2)	21 (38.2)	51 (19.0)	0.004
Albumin, <3.5 mg/dL	30 (9.3)	11 (20.0)	19 (7.1)	0.008
C-reactive protein, >0.5 mg/dL	152 (53.1)	41 (74.5)	111 (41.3)	<0.001
Treatments				
Lopinavir/Ritonavir	175 (54.0)	40 (72.7)	135 (50.2)	0.003
Chloroquine	54 (16.7)	17 (30.9)	37 (13.8)	0.005
Antibiotics	142 (43.8)	40 (72.7)	102 (37.9)	<0.001
Macrolide	86 (26.5)	23 (41.8)	63 (23.4)	0.007
Pegylated interferon-α	56 (17.3)	16 (29.1)	40 (14.9)	0.018
O2 supplement	61 (18.8)	21 (38.2)	40 (14.9)	<0.001

Results are presented as means ± SDs or as frequency (%). *p*-values are for differences between the diabetes and non-diabetes groups.

**Table 3 healthcare-08-00338-t003:** Univariate regression analysis results for risk factors of mortality in COVID-19 patients with diabetes ^a^.

Variables	OR (95%CI)	*p*-Value
Age group, years		
≥70	23.882 (6.914–82.490)	<0.001
Body Mass Index, kg/m^2^	0.818 (0.658–1.017)	0.071
Systolic blood pressure, mmHg	0.989 (0.959–1.019)	0.458
Diastolic blood pressure, mmHg	0.984 (0.938–1.033)	0.518
Body temperature, ≥37.5 ℃	3.037 (0.441–20.907)	0.259
Sex		
Male	0.948 (0.130–6.922)	0.958
Female		
Current smoker	3.815 (0.376–38.721)	0.258
Alcohol drinker	0.931 (0.075–11.483)	0.955
Resident in care facility	0.572 (0.087–3.766)	0.572
Comorbidity		
Hypertension	0.415 (0.059–2.902)	0.415
Cardiovascular disease	0.895 (0.068–11.744)	0.933
Cerebrovascular disease	0.954 (0.123–7.393)	0.954
Dementia	0.683 (0.109–4.280)	0.684
Malignancy	1.036 (0.056–19.182)	0.981
Symptoms on admission		
Dyspnea	1.736 (0.267–11.299)	0.564
Febrile sense	1.689 (0.325–8.764)	0.533
Diarrhea	1.509 (0.075–30.184)	0.788
Chest imaging		
Bilateral infiltration	4.682 (0.429–51.019)	0.205
Laboratory findings		
Blood glucose	0.996 (0.985–1.008)	0.547
HbA1c	0.892 (0.351–2.265)	0.810
Leukocyte, >10.7 × 10^9^/L	16.319 (0.876–303.999)	0.061
Lymphocyte, <1.0 × 10^9^/L	4.375 (0.800–23.937)	0.089
High Aspartate aminotransferase	2.888 (0.490–17.014)	0.241
High Lactate dehydrogenase	13.266 (1.881–93.545)	0.009
High Creatinine	1.807 (0.216–15.115)	0.585
Anemia	2.862 (0.579–14.143)	0.197
Albumin, <3.5 mg/dL	2.464 (0.426–14.261)	0.314
C-reactive protein	2.004 (0.604–16.946)	0.256

^a^ Adjusted for age. OR, odds ratio. CI, confidence interval.

**Table 4 healthcare-08-00338-t004:** Multivariate regression analysis results for risk factors of mortality in COVID-19 patients with diabetes.

Variables	OR (95% CI)	*p*-Value
≥70, year	17.415 (1.797–168.791)	0.014
High Lactate dehydrogenase	9.703 (1.812–51.961)	0.008

OR, odds ratio. CI, confidence interval.

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
