# Peer review of "Mortality Rate and Predictors of Mortality in Hospitalized COVID-19 Patients with Diabetes"

_healthcare, 2020, doi:10.3390/healthcare8030338_

Round 1

Reviewer 1 Report

This is a cross-sectional study of people hospitalized for COVID-19, including 55 with diabetes. The findings indicate a much higher mortality rate for people with diabetes, especially older and with high LDH. These are significant findings that may provide some novel clinical info to help us better treat diabetics with COVID. One major limitation is the small number of patients with diabetes included (n=55). The findings are a bit limited because of the narrow scope of the analyses and small sample size, but the high relevance make this article important for readers at this time.

No major issues

Minor issues.

Line 22-23. No need to provide abbreviations here for DUGH and AMC

Line 31. Remove 'older' since that is a result explained at the end of the sentence (>70 yrs)

Are these all type II diabetes patients? If so, that should be clarified at least in the methods

Line 39. Replace ON July 28 with AS OF July 28.

Line 42. Replace AND with AND/OR. Remove 's' from appears

Line 52. Change meta-analysis to meta-analyses

Line 70. Remove second parenthasis around (CT))

Line 100. Add period after '55 had diabetes.', then start new sentence. Same with Line 105-106 after (67.3% vs. 43.1%)

Tables 1 & 2. Instead of 'number', use 'frequency' in the legend at the bottom.

Lines 131-133. Please remove the instructional text.

Line 134. I'm unclear what the 'management of mortality' means? Do you mean disease management?

Line 137. I thnk the sentence, 'This pandemic has adequately proven...' is too strongly worded. Please revise to leave some room for developing scientific consensus.

Line 148. Put the period on the outside of [25,26].

Line 171. replace 'have been' with 'be'

Author Response

Journal: Healthcare

Manuscript ID: healthcare-934752

Title: Mortality Rate and Predictors of Mortality in Hospitalized COVID-19 Patients with Diabetes

First Reviewer-First Round

General Comments and Suggestions for Authors

This is a cross-sectional study of people hospitalized for COVID-19, including 55 with diabetes. The findings indicate a much higher mortality rate for people with diabetes, especially older and with high LDH. These are significant findings that may provide some novel clinical info to help us better treat diabetics with COVID. One major limitation is the small number of patients with diabetes included (n=55). The findings are a bit limited because of the narrow scope of the analyses and small sample size, but the high relevance make this article important for readers at this time.

No major issues

Response: We thank you very much and highly appreciate reviewer’s valuable comments and suggestions. We have revised the manuscript based on reviewers’ comments and suggestions. The issue of sample size has been discussed asone of the limitations of the study. All changes are marked with blue colored writing in this revised version of the manuscript to allow reviewers’ verifications.

Minor issues.

Line 22-23. No need to provide abbreviations here for DUGH and AMC

Response: Agree. The sentence is restructured.

Line 31. Remove 'older' since that is a result explained at the end of the sentence (>70 yrs)

Response: Agree. Corrected.

Are these all type II diabetes patients? If so, that should be clarified at least in the methods

Response: Agree. Corrected as suggested.

Line 39. Replace ON July 28 with AS OF July 28.

Response: Agree. Corrected.

Line 42. Replace AND with AND/OR. Remove 's' from appears

Response: Agree. Corrected.

Line 52. Change meta-analysis to meta-analyses

Response: Agree. Corrected.

Line 70. Remove second parenthasis around (CT))

Response: Agree. Corrected.

Line 100. Add period after '55 had diabetes.', then start new sentence. Same with Line 105-106 after (67.3% vs. 43.1%)

Response: Agree. Corrected.

Tables 1 & 2. Instead of 'number', use 'frequency' in the legend at the bottom.

Response: Agree. Corrected.

Lines 131-133. Please remove the instructional text.

Response: Agree. The suggested lines were removed.

Line 134. I'm unclear what the 'management of mortality' means? Do you mean disease management?

Response: Agree. Sentence restructured to make it readable and meaningful.

Line 137. I think the sentence, 'This pandemic has adequately proven...' is too strongly worded. Please revise to leave some room for developing scientific consensus.

Response:  Strongly agree. Sentence restructured to make it readable and meaningful.

Line 148. Put the period on the outside of [25,26].

Response: Agree. Corrected.

Line 171. replace 'have been' with 'be'

Response: Agree. Replaced as suggested.

Reviewer 2 Report

COVID-19 pandemic threatens human health, daily life, and economics. Particular populations are vulnerable to COVID-19 suffering. A lack of vaccination and effective treatment highlight an importance to identify susceptible subjects and design any preventive strategies against COVID-19 mortality. In this cross-sectional study, COVID-19 patients with diabetes suffered from increased mortality, particularly those are older than 70 years and higher serum LDH level.

There were many studies identified risk factors and comorbid conditions of COVID-19 mortality. Age, diabetes, cardiovascular diseases, hyperglycemia, HbA1c, CRP, and LDH. From the enrolled 324 patients dividing to diabetes (55) and non-diabetes (269), regression analysis of mortality in COVID-19 patients with diabetes revealed poor predictors, age than 70 years and high serum LDH. The identified predictors all had been reported in previous studies. Thus, the analysis and proposed strategies should be in depth.

  1. Age is a risk factor of COVID-19 mortality. The age 70 should have discussion with other studies.
  2. The value of LDH≥345IU/L has been set as a factor of COVID-19 mortality. Current study, the cut-off value was set at 225 and 214 for male and female, respectively. A comparison and discussion is essential.
  3. Prospective strategies, implications, and directions should be proposed and discussed to extend current findings.

Author Response

Journal: Healthcare

Manuscript ID: healthcare-934752

Title: Mortality Rate and Predictors of Mortality in Hospitalized COVID-19 Patients with Diabetes

Reviewer 2-First Round

Comments and Suggestions for Authors

COVID-19 pandemic threatens human health, daily life, and economics. Particular populations are vulnerable to COVID-19 suffering. A lack of vaccination and effective treatment highlight an importance to identify susceptible subjects and design any preventive strategies against COVID-19 mortality. In this cross-sectional study, COVID-19 patients with diabetes suffered from increased mortality, particularly those are older than 70 years and higher serum LDH level.

 There were many studies identified risk factors and comorbid conditions of COVID-19 mortality. Age, diabetes, cardiovascular diseases, hyperglycemia, HbA1c, CRP, and LDH. From the enrolled 324 patients dividing to diabetes (55) and non-diabetes (269), regression analysis of mortality in COVID-19 patients with diabetes revealed poor predictors, age than 70 years and high serum LDH. The identified predictors all had been reported in previous studies. Thus, the analysis and proposed strategies should be in depth.

Response: Thank you very much for the fruitful comments and suggestions. We have revised the manuscript based on reviewers’ comments and suggestions. We agree the reviewer’s comment. However, it is far less common study in Korean human subjects. All changes are marked with blue colored writing in this revised version of the manuscript to allow reviewers’ verifications.

Comment: Age is a risk factor of COVID-19 mortality. The age 70 should have discussion with other studies.

Response: Agree. We appreciate the reviewer’s comment. We have added the discussion section in this regard

  1. The value of LDH≥345IU/L has been set as a factor of COVID-19 mortality. Current study, the cut-off value was set at 225 and 214 for male and female, respectively. A comparison and discussion is essential.

Response: Agree. We have discussed shortly about the variability of cut-off measurements shortly as suggested.

2. Prospective strategies, implications, and directions should be proposed and discussed to extend current findings.

Response: Agree. We have changed to some extent in discussion and conclusions as suggested.

Reviewer 3 Report

In my opinion, the manuscript is well written and reads correctly.

The studies presented are easily understood by the reader.

However, I consider that a study of these characteristics would imply contributing a higher number of patients, n = 324, which I find few.

And The Conclusions seem to me to be few. In my opinion, the authors could be extended a little more with the amount of results that they express and be able to demonstrate the quality of their studies in their manuscript.

For these reasons I recommend: Accept after minor revision

Author Response

Journal: Healthcare

Manuscript ID: healthcare-934752

Title: Mortality Rate and Predictors of Mortality in Hospitalized COVID-19 Patients with Diabetes

Reviewer 3-First Round

Comments and Suggestions for Authors

In my opinion, the manuscript is well written and reads correctly.

The studies presented are easily understood by the reader.

However, I consider that a study of these characteristics would imply contributing a higher number of patients, n = 324, which I find few.And the Conclusions seem to me to be few. In my opinion, the authors could be extended a little more with the amount of results that they express and be able to demonstrate the quality of their studies in their manuscript.

For these reasons I recommend: Accept after minor revision.

Response: Thank you very much for your encouragement. We have revised the manuscript based on reviewers’ comments and suggestions. Regarding the issue of sample size, we have mentioned the issue as one the limitation of this study. In addition, we have added the conclusion section as per the reviewers’ suggestions. All changes are marked with blue colored writing in this revised version of the manuscript to allow reviewers’ verifications.

Round 2

Reviewer 2 Report

There is no additional comment.